# NDRG2 Sensitizes Myeloid Leukemia to Arsenic Trioxide via GSK3β–NDRG2–PP2A Complex Formation

**DOI:** 10.3390/cells8050495

**Published:** 2019-05-22

**Authors:** Soojong Park, Hyun-Tak Han, Sang-Seok Oh, Dong Hyeok Kim, Jin-Woo Jeong, Ki Won Lee, Minju Kim, Jong Seok Lim, Yong Yeon Cho, Cheol Hwangbo, Jiyun Yoo, Kwang Dong Kim

**Affiliations:** 1Division of Applied Life Science (BK21 Plus), Gyeongsang National University, Jinju 52828, Korea; soojongpark@kribb.re.kr (S.P.); entreluzyluz@naver.com (H.-T.H.); leemaskup@naver.com (K.W.L.) mong2nuna@gnu.ac.kr (M.K.); yooj@gnu.ac.kr (J.Y.); 2Gene & Cell Therapy Team, Division of Drug Development & Optimization, New Drug Development Center, Osong Medical Innovation Foundation, Osongsaengmyung-ro 123, Osong-eup, Heungdeok-gu, Cheongju-si 28160, Chungbuk, Korea; ssoh@kbiohealth.kr; 3Division of bacterial diseases, Korea Centers for Disease and Control, Prevention, Osong-eup 28159, Korea; kimdonghyeok@korea.kr; 4Freshwater Bioresources Utilization Bureau, Nakdonggang National Institute of Biological Resources, Sangju 37242, Korea; jwjeong@nnibr.re.kr; 5Department of Biological Sciences and the Research Center for Women’s Disease, Sookmyung Women’s University, Seoul 04310, Korea; jslim@sookmyung.ac.kr; 6Integrated Research Institute of Pharmaceutical Sciences, BK21 PLUS Team & BRL, College of Pharmacy, The Catholic University of Korea, Wonmi-gu, Bucheon-si, 14662, Korea; yongyeon@catholic.ac.kr; 7Division of Life Science, Gyeongsang National University, Jinju 52828, Korea; chwangbo@gnu.ac.kr; 8Plant Molecular Biology and Biotechnology Research Center (PMBBRC), Gyeongsang National University, Jinju 52828, Korea

**Keywords:** PP2A–NDRG2–GSK3β complex, myeloid leukemia, U937, Mcl-1, apoptosis, arsenic trioxide

## Abstract

N-Myc downstream-regulated gene 2 (NDRG2) was characterized as a tumor suppressor, inducing anti-metastatic and anti-proliferative effects in several tumor cells. However, NDRG2 functions on anticancer drug sensitivity, and its molecular mechanisms are yet to be fully investigated. In this study, we investigated the mechanism of NDRG2-induced sensitization to As_2_O_3_ in the U937 cell line, which is one of the most frequently used cells in the field of resistance to As_2_O_3_. NDRG2-overexpressing U937 cells (U937-NDRG2) showed a higher sensitivity to As_2_O_3_ than mock control U937 cell (U937-Mock). The higher sensitivity to As_2_O_3_ in U937-NDRG2 was associated with Mcl-1 degradation through glycogen synthase kinase 3β (GSK3β) activation. Inhibitory phosphorylation of GSK3β was significantly reduced in U937-NDRG2, and the reduction was diminished by okadaic acid, a protein phosphatase inhibitor. NDRG2 mediated the interaction between GSK3β and protein phosphatase 2A (PP2A), inducing dephosphorylation of GSK3β at S9 by PP2A. Although the C-terminal deletion mutant of NDRG2 (ΔC NDRG2), which could not interact with PP2A, interacted with GSK3β, the mutant failed to dephosphorylate GSK3β at S9 and increased sensitivity to As_2_O_3_. Our findings suggest that NDRG2 is a kind of adaptor protein mediating the interaction between GSK3β and PP2A, inducing GSK3β activation through dephosphorylation at S9 by PP2A, which increases sensitivity to As_2_O_3_ in U937 cells.

## 1. Introduction

N-Myc downstream-regulated gene 2 (NDRG2) plays an important role in tumor suppression in several cancers or cancer cell lines [1,2]. NDRG2 expression is positively correlated with tumor differentiation but negatively correlated with metastasis and TNM classification of malignant tumors (TNM) stage [3,4,5,6]. Signaling pathways inducing epithelial–mesenchymal transition (EMT), including Wnt and phosphoinositide 3-kinase (PI3K), were regulated by glycogen synthase kinase 3β (GSK3β) [7,8]. NDRG2 expression was positively correlated with GSK3β activation [9,10], and various NDRG2-mediated GSK3β regulation mechanisms were suggested [11,12]. NDRG2 recruits PP2A to dephosphorylate phosphatase and tensin homolog (PTEN), which inhibits protein kinase B (AKT) activity through dephosphorylation. Active AKT induces phosphorylation of GSK3β at S9, an inactive form of GSK3β. Therefore, the PTEN–NDRG2–PP2A complex might induce GSK3β activation through inhibition of AKT activity, which inhibits tumor metastasis. 

As_2_O_3_ is an effective anticancer drug for acute promyelocytic leukemia (APL) patients [13,14,15]. However, the drug’s clinical application is limited due to low sensitivity in other types of leukemia and its side effects [16,17,18]. It is known that Mcl-1 contributes to protect cells from apoptosis in APL cells [19]. The PI3K/AKT/mTOR pathway is known to promote cell survival through translational control of Mcl-1 in acute myeloid leukemia (AML) [20]. Mcl-1 has a short half-life due to rapid degradation after phosphorylation by GSK3β which is negatively regulated by active AKT [21]. Actually, Mcl-1 protein levels were decreased in As_2_O_3_-treated NB4 cells and As_2_O_3_-sensitive APL, and the decrease was dependent on GSK3β activation [22]. Therefore, strategies for AKT inhibition, GSK3β activation, and degradation of Mcl-1 could be important points to improve the sensitivity to As_2_O_3_ in APL.

In this study, we used U937 cells, which do not express NDRG2, to investigate whether NDRG2 affects cancer drug sensitivity, because the acute myeloid leukemia cell line, U937, is relatively resistant to As_2_O_3_ [23,24]. We found that NDRG2-overexpressing U937 (U937-NDRG2) cells were more sensitive to As_2_O_3_ compared to parental U937 (U937-Mock) cells. Although NDRG2 negatively regulates AKT activity through the PTEN–NDRG2–PP2A complex, the sensitivity of U937-NDRG2 cells to As_2_O_3_ does not mean that the complex inhibits the PI3K/AKT pathway, since the U937 cell line possesses a PTEN frameshift mutant [25]. In this study, we investigated and identified a novel mechanism of NDRG2-mediated GSK3β activation, which enhanced the sensitivity of U937 to As_2_O_3_ through Mcl-1 degradation. 

## 2. Materials and Methods

### 2.1. Cell Culture, Reagents, and Short Hairpin RNAs (shRNAs)

Immortalized embryonic kidney cell lines, 293T/17 and HEK293, were purchased from American Type Culture Collection, ATCC (Manassas, VA, USA). The U937, U937-Mock, and U937-NDRG2 cell lines [26] were cultured in Roswell Park Memorial Institute (RPMI) -1640 medium containing 10% fetal bovine serum (FBS) (ATCC, Manassas, VA, USA), HEPES (Lonza, Basel, Swiss), β-mercaptoethanol (Sigma-Aldrich, St. Louis, MO, USA), and penicillin and streptomycin (Lonza, Basel, Swiss). The 293T/17 and HEK293 cells were cultured in Dulbecco’s modified Eagle’s medium containing 10% FBS, penicillin, and streptomycin at 37 °C in 5% CO_2_. In some experiments, the cells were treated with chemicals As_2_O_3_, MG132, 2’,7’-dichlorofluorescin diacetate (DCFH-DA), okadaic acid, SB216763 (Sigma-Aldrich, MO, USA), and z-VAD-fmk (ENZO, Seoul, Korea). The shMcl-1 clones (TRCN0000005514 and TRCN0000005516) were purchased from Sigma-Aldrich.

### 2.2. Western Blotting

The cell lysates were prepared in ProNA^TM^ CETi Lysis Buffer (TransLab, Daejeon, Korea). The protein extracts were concentrated by the Bradford protein assay (Bio-Rad, Hercules, CA, USA) according to the manufacturer’s instructions. Equal amounts of proteins were separated by SDS-PAGE and transferred onto a polyvinylidene difluoride (PVDF) membrane. Membrane blocking was performed using 5% skimmed milk (in Tris Buffered Saline with Tween 20) for 1 h at room temperature (RT), and the membrane was incubated with the primary antibody at 4 °C overnight. The horseradish peroxidase (HRP)-conjugated antibodies were then treated at RT for 1 h. The HRP signal was activated using Clarity™ ECL Western Blotting Substrate (Bio-Rad, Hercules, CA, USA). Band intensity was calculated using the Image J software (1.50g version, NIH, Bethesda, MD, USA). B-cell lymphoma 2 (Bcl-2), B-cell lymphoma–extra large (Bcl-xL), Bcl-2-associated X protein (Bax), and NDRG2 antibodies were purchased from Santa Cruz Biotechnology (TX, USA). Antibodies for PP2Ac, phosphorylated (p)-GSK3β (Ser9), GSK3β, Mcl-1, caspase-3, caspase-9, poly-ADP ribose polymerase (PARP), p-AKT (Thr308), p-AKT (Ser473), AKT, GST, and mouse anti-rabbit IgG-HRP were purchased from Cell Signaling Technology (Danvers, MA, USA). Anti-α-tubulin antibody was purchased from Sigma-Aldrich (St. Louis, MO, USA). Anti-mouse Ig HRP was purchased from eBioscience (San Diego, CA, USA), and hemagglutinin (HA) antibody was purchased from abm (abm^®^, Vancouver, Canada). 

### 2.3. Co-Immunoprecipitation and Glutathion S-Tranferase Tag(GST) Pull-Down Assay

After transfecting relevant vectors into HEK293 cells, or treatment of U937 cells with As_2_O_3_, the cells were harvested and lysed with buffer containing 0.5% NP40, 50 mM Tris-HCl (pH 8.0), 150 mM NaCl, a protease inhibitor cocktail, and a phosphatase inhibitor cocktail. For GST pull-down assays, the cellular supernatants were incubated with 30 μL of 50% glutathione sepharose 4B slurry (GE Healthcare, Philadelphia, PA, USA) at 4 °C overnight. After incubation, the beads were washed three times using buffer containing 0.5% NP40, 20 mM Tris-HCl (pH 8.0), 100 mM NaCl, and 1 mM ethylenediaminetetraacetic acid (EDTA) for 10 min. For co-immunoprecipitation, the cellular supernatants were incubated with 2 μg Flag antibody at 4 °C overnight, after which they were incubated with 20 μL of Protein G Plus/Protein A agarose beads (Millipore, Darmstadt, Germany) for an additional 2 h at 4 °C. The beads were washed three times, using the aforementioned buffer, for 10 min. All washed beads were incubated with protein sample buffer and boiled for 5 min. Twenty micrograms of protein from the cellular supernatants was used as whole-cell lysate (WCL) for downstream evaluation.

### 2.4. PP2Ac Activity Assay

PP2A activity was assessed using the Human/Mouse/Rat Active PP2A DuoSet IC Activity Assay (R&D systems, Minneapolis, MN, USA) according to the manufacturer’s instructions. The day before cell harvest, 96-well microplates were treated with PP2A-capturing antibody. The wells were then washed using buffer (0.05% Tween-20 in phosphate-buffered saline, pH 7.2–7.4) and blocked using 1% bovine serum albumin (BSA) and 0.05% NaN_3_ in PBS, pH 7.2–7.4. The conditioned cells were lysed with lysis buffer #8 (50 mM HEPES, 0.1 mM ethylene glycol tetraacetic acid (EGTA), 0.1 mM EDTA, 120 mM NaCl, 0.5% NP40 alternative (pH 7.5), and a protease cocktail). Protein samples (240 μg) were added to the coated 96-well microplates. The wells containing captured PP2A were treated with Ser/Thr phosphatase substrates, malachite green reagent A, and malachite green reagent B. Absorbance at 590 nm was measured using a Bio-Rad microplate reader (Bio-Rad, Hercules, CA, USA).

### 2.5. Lentivirus Production and Infection

For Mcl-1 knockdown, validated shRNAs against the gene were purchased from Sigma-Aldrich, and viral production and infection of cell lines were performed according to the ViraPower^TM^ Lentiviral Expression System protocol (Invitrogen, Carlsbad, CA, USA). The supernatant was used to infect U937 cells in a 12-well plate, and the infected cell population was selected by incubating with 5 μg/mL puromycin overnight.

### 2.6. Flow Cytometry

For apoptosis analysis, the cells were stained with 1× Annexin V binding buffer containing Annexin V-fluorescein isothiocyanate (BD Bioscience, Franklin Lakes, NJ, USA) and propidium iodide (PI; Sigma-Aldrich). For mitochondrial potential analysis, the cells were stained with 250 nM Mitotracker CMXRos (Invitrogen, Carlsbad, CA, USA) for 30 min at 37 °C. Fluorescence was measured with a FACSVerse™ flow cytometer (BD Bioscience, Franklin Lakes, NJ, USA) and analyzed using FlowJo V10 software (FlowJo, Ashland, OR, USA).

### 2.7. Statistics

Data were acquired from three independent experiments and analyzed using the unpaired Student’s *t*-test. A *p*-value < 0.05 was considered to be statistically significant.

## 3. Results

### 3.1. NDRG2 Expression Sensitizes U937 Cells to As_2_O_3_

To determine whether NDRG2 expression affects sensitivity to As_2_O_3_ in U937 cells, three NDRG2-overexpressing cell lines (U937-NDRG2) were established. When treated with 2 μM As_2_O_3_, higher apoptotic rates were observed in the U937-NDRG2 cells compared with U937-Mock cells (Figure 1A). U937-NDRG2 clone 1 was used in all experiments below. The increased apoptotic rate in the U937-NDRG2 cells also tended to show time or dose dependency (Figure 1B), and could be inhibited by a pan-caspase inhibitor, zVad-fmk (Figure 1C). Additionally, cleaved levels of caspase-3 and PARP after As_2_O_3_ treatment were higher in the U937-NDRG2 than in the U937-Mock cells (Figure 1D). These results suggest that NDRG2 increases the sensitivity of U937 to As_2_O_3_-induced apoptosis in a caspase-dependent manner.

### 3.2. The Sensitivity of U937-NDRG2 to As_2_O_3_ Is Determined by Mcl-1 Degradation 

Among Bcl-2 family proteins, Mcl-1, an anti-apoptotic protein, was significantly decreased in As_2_O_3_-treated U937-NDRG2 cells (Figure 2A). For validating whether the decrease of Mcl-1 was critical for the observed increased sensitivity to As_2_O_3_, Mcl-1 expression was silenced in the U937-Mock cells using shMcl-1. These cells showed higher sensitivity to As_2_O_3_, similar to the U937-NDRG2 cells (Figure 2B). Reduced Mcl-1 protein was not regulated at the transcriptional level (Figure 2C), but rather due to decreased protein stability, which could be offset by a proteasome complex inhibitor, MG132 (Figure 2D). Taken together, these results indicate that NDRG2 overexpression increases the sensitivity of the cells to As_2_O_3_ by reducing Mcl-1 protein stability. 

### 3.3. As_2_O_3_ Induces Mcl-1 Degradation through GSK3β Activation in U937-NDRG2

Active GSK3β phosphorylates Mcl-1, which is then degraded by the ubiquitin proteasome system [21]. In this study, we found that GSK3β was rapidly activated in U937-NDRG2 cells (Figure 3A). To confirm whether active GSK3β was associated with the sensitivity of U937-NDRG2 to As_2_O_3_, the U937-NDRG2 cells were co-treated with As_2_O_3_ and SB216763, a GSK3β inhibitor. The inhibition of GSK3β completely blocked As_2_O_3_-induced apoptosis and depolarization of mitochondria in these cells (Figure 3B,C). Additionally, As_2_O_3_-induced Mcl-1 degradation and caspase-3 cleavage were completely inhibited by SB216763 (Figure 3D). These results suggest that As_2_O_3_-induced GSK3β activation in the NDRG2 overexpressed condition increases the sensitivity of the cells to As_2_O_3_ through Mcl-1 degradation.

### 3.4. NDRG2 Mediates the Interaction between GSK3β and PP2A

Although a report suggested that NDRG2 mediates PP2A–PTEN complex formation, which inhibits the PI3K/AKT pathway through PTEN activation by PP2A [11], the phosphorylation on T308 of AKT regulated by PTEN was not changed in U937-Mock and U937-NDRG2 cells (Figure 4A). To determine whether PP2A could activate GSK3β, the U937-NDRG2 cells were treated with As_2_O_3_ in the presence or absence of okadaic acid, a broad PP2A inhibitor. PP2A inhibition rescued the inhibitory phosphorylation of GSK3β, which blocked the degradation of Mcl-1 (Figure 4B). NDRG2 might increase PP2A expression levels or its activity in the context of As_2_O_3_ treatment; however, neither were affected in the U937-NDRG2 cells (Figure 4C). To discern whether interaction among GSK3β, NDRG2, and PP2A was associated with the increased sensitivity to As_2_O_3_, we firstly performed GST pull-down assays with cell lysate from HEK293 cells transiently expressing GST or GST–NDRG2, and HA–GSK3β. GST–NDRG2 interacted with endogenous PP2Ac and exogenous HA–GSK3β, while GST did not interact with them (Figure 4D). Inversely, mock vector or HA–GSK3β was co-transfected with GST–NDRG2, and immunoprecipitation was performed with an anti-HA antibody. HA–GSK3β successfully interacted with exogenous GST–NDRG2 and endogenous PP2Ac (Figure 4E). To investigate whether NDRG2 plays a role as a bridge mediating the interaction between GSK3β and PP2A, the interaction between HA–GSK3β and endogenous PP2A was validated in the presence or absence of GST–NDRG2. HA–GSK3β was precipitated with endogenous PP2A, but only in the presence of GST–NDRG2 (Figure 4F). These results suggest that NDRG2 mediates the interaction between GSK3β and PP2Ac.

### 3.5. The Formation of GSK3β/NDRG2/PP2A Complex Determines the Sensitivity of U937 to As_2_O_3_

A previous study reported that PP2Ac interacts with NDRG2 through its C-terminal domain [11]. Therefore, we established a U937 cell line expressing the C-terminal deletion mutant of NDRG2 (U937-ΔC NDRG2). Although ΔC NDRG2 could not interact with PP2A, it could still maintain its interaction with GSK3β (Figure 5A). As_2_O_3_-induced GSK3β activation and Mcl-1 degradation was not observed in the U937-ΔC NDRG2 cells (Figure 5B). Furthermore, the higher sensitivity to As_2_O_3_ shown in U937-NDRG2 cells was abolished in the U937-ΔC NDRG2 cells (Figure 5C). Therefore, the formation of GSK3β/NDRG2/PP2A complex was necessary for GSK3β activation and subsequent Mcl-1 degradation in U937-NDRG2 treated with As_2_O_3_.

## 4. Discussion

NDRG2, as a tumor suppressor, mainly suppresses cancer development and progression. It was proposed that, in clinical investigations, NDRG2 is positively correlated with survival rate and disease-free survival (DFS) probability, and negatively correlated with lymph node metastasis and TNM stage [4,5,6]. In this study, we investigated the molecular mechanism of NDRG2 function, as a kind of tumor suppressive gene, to overcome the low chemosensitivity of tumor cells.

As_2_O_3_ is approved by the Food and Drug Administration (FDA) to treat primary or relapsed acute promyelocytic leukemia (APL), a subtype of acute myeloid leukemia (AML) [27]. The therapeutic potential of As_2_O_3_ is not restricted to APL cells, and its application can induce apoptosis in non-APL acute myeloid leukemia cells, chronic myeloid leukemia cells, and other solid tumors in vitro [28,29,30]. To investigate NDRG2 function associated with drug sensitivity, the U937 cell line was used, because the cell line does not express NDRG2 and it is a representative one showing very low sensitivity to As_2_O_3_. We established NDRG2-overexpressing U937 (U937-NDRG2) cell lines, and the cells showed higher sensitivity to As_2_O_3_ compared with U937-Mock cells (Figure 1). The higher sensitivity was due to Mcl-1 degradation (Figure 2). Actually, the downregulation of Mcl-1 through GSK3β activation contributed to As_2_O_3_-induced apoptosis in acute myeloid leukemia [22]. The primary kinase regulating Mcl-1 stability is GSK3β, which phosphorylates Mcl-1 at S155, S159, and T163 [31,32]. The phosphorylated Mcl-1 is ubiquitinated by E3 ligases, F-box/WD repeat-containing protein 1A (β-TrCP), Mcl-1 ubiquitin ligase (Mule), or F-box/WD repeat-containing protein 7 (FBW7), and undergoes proteasome-dependent degradation [32,33,34]. Effective GSK3β activation and Mcl-1 degradation were induced in As_2_O_3_-treated U937-NDRG2 cells, and the inhibition of GSK3β using a specific inhibitor, SB216763, effectively decreased the sensitivity of the cells to As_2_O_3_, as well as Mcl-1 degradation (Figure 3). Mcl-1 is known as a crucial component in As_2_O_3_-induced apoptosis through GSK3β activation in acute myeloid leukemia [22,35].

As an upstream kinase of GSK3β, AKT is directly associated with the phosphorylation of GSK3β on Ser9, and its oncogenic mutations driving over-activation of PI3K/AKT pathway tend to result in excessive inactivation of GSK3β in various cancer cell lines [36]. Recently, NDRG2 was shown to inhibit PI3K/AKT signaling by activating PTEN through the recruitment of PP2A [11]. Furthermore, NDRG2-deficient mice showed inhibition of GSK3β through activated PI3K/AKT signaling [12]. In our study, although we observed GSK3β activation and Mcl-1 degradation in U937-NDRG2 treated with As_2_O_3_, these conditions did not reduce phosphorylation of T308 in AKT (Figure 4A). Therefore, this result suggested that the PI3K/AKT signaling regulated by PTEN/NDRG2/PP2A was not involved in the sensitivity of U937-NDRG2 to As_2_O_3_. Furthermore, since PTEN is mutated in the U937 cell line [37], the mechanism involving the inhibition of AKT by PTEN followed by GSK3β activation could be ruled out. A report suggested that PP2A directly dephosphorylates GSK3β through the relay of DNAJ homolog subfamily B member 6 (DNAJB6) [38]. DNAJB6 binds HSPA8 (heat-shock cognate protein, HSC70) and causes dephosphorylation of GSK3β at Ser9 by recruiting protein phosphatase PP2A. In this study, we hypothesize that NDRG2 acts as a bridge connecting GSK3β and PP2A, so that PP2A dephosphorylates the inhibitory phosphorylation of GSK3β. As shown in Figure 4, NDRG2 protein interacts with GSK3β and PP2A. Moreover, GSK3β and PP2A could not interact in the absence of NDRG2 expression. Although C-terminal-deleted NDRG2, which cannot bind to PP2A, interacts with GSK3β, GSK3β activation, Mcl-1 degradation, and higher sensitivity to As_2_O_3_ were suppressed in U937-ΔC NDRG2 cells (Figure 5). Although the NDRG2-mediated complex formation, GSK3β/NDRG2/PP2A, could not activate PP2A directly, it might be possible that As_2_O_3_ activates PP2A [39] and then accelerates activation of GSK3β through the complex.

Finally, our findings suggest a new regulatory AKT-independent pathway on GSK3β activation, in which NDRG2 positively regulates GSK3β by forming the GSK3β/NDRG2/PP2Acomplex, leading to PP2A-mediated removal of inhibitory phosphate on GSK3β (Figure 6). Thus, the GSK3β/NDRG2/PP2A complex is necessary for the GSK3β-mediated Mcl-1 degradation pathway, which consequently leads to enhanced sensitivity of the U937 cells to As_2_O_3_. 

## Figures and Tables

**Figure 1 cells-08-00495-f001:**
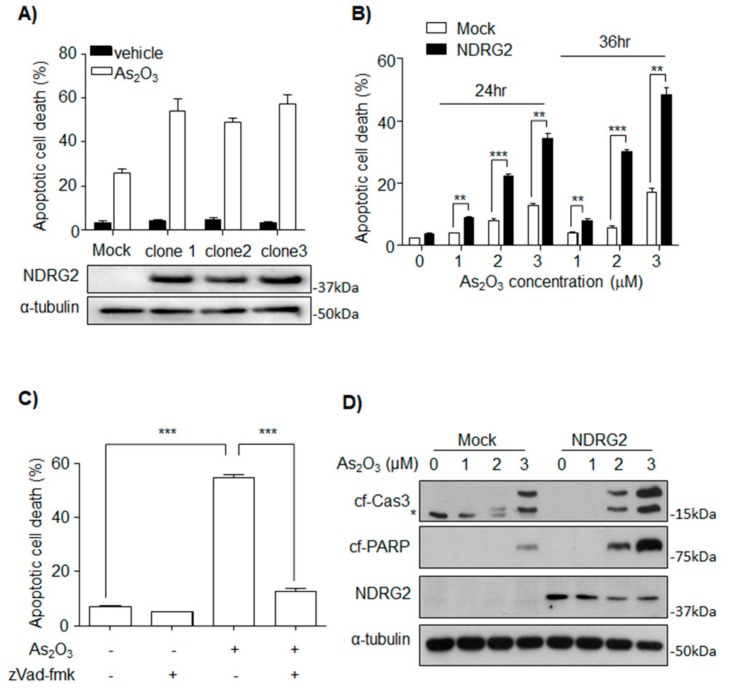
N-Myc downstream-regulated gene 2 (NDRG2) overexpression sensitized U937 cells to As_2_O_3_ in a caspase-dependent manner. (**A**) U937-Mock and three U937-NDRG2 lines were incubated with As_2_O_3_ (2 μM, 24 h). The cells were stained with Annexin V/propidium iodide (PI) and analyzed by flow cytometry. (**B**) The cells were incubated with As_2_O_3_ at the indicated time and concentration, and apoptotic cell population was validated with Annexin V/PI staining. (**C**) Here, 2 μM As_2_O_3_ was treated in the presence or absence of zVad-fmk (Pan-caspase inhibitor, 50 μM). The apoptotic population was validated with Annexin V/PI staining. (**D**) U937-Mock and U937-NDRG2 cells treated with As_2_O_3_ at the indicated dose. The cleavages (Cf) of Caspase3 and PARP were analyzed by immunoblotting. * is non-specific band. ** *p* < 0.01, *** *p* < 0.005 determined from *t*-tests. Data are presented as means ± standard error of the mean (SEM).

**Figure 2 cells-08-00495-f002:**
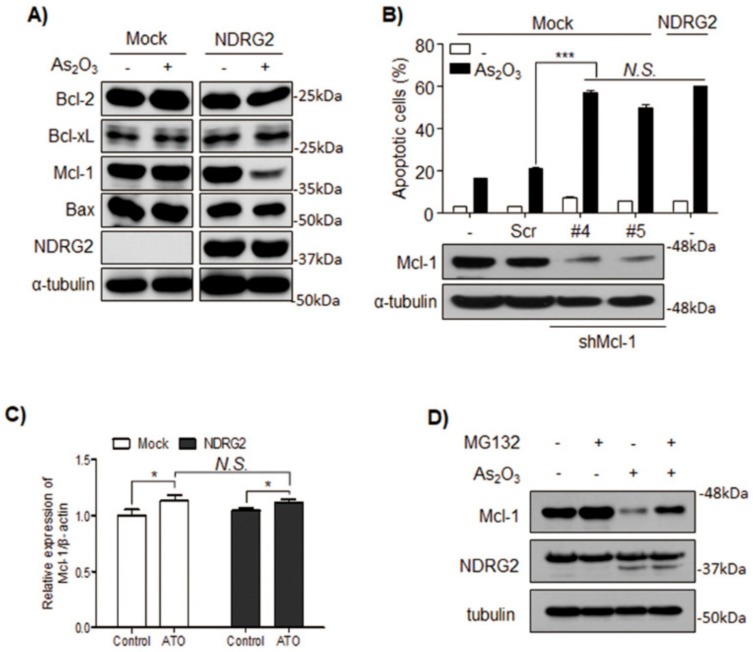
The sensitivity of U937-NDRG2 to As_2_O_3_ is mediated through Mcl-1 degradation. (**A**) Expression levels of pro-apoptotic Bcl-2 family protein (Bax) and pro-survival Bcl-2 family protein were analyzed by immunoblotting. (**B**) U937-Mock was infected by a lentivirus expressing scr or short hairpin Mcl-1 (shMcl-1) (clone #4 and #5). Successful knockdown of Mcl-1 was confirmed by immunoblotting, and apoptotic rate was analyzed using flow cytometry. (**C**) Mcl-1 messenger RNA (mRNA) expression in U937-Mock and U937-NDRG2 in the presence or absence of As_2_O_3_ was validated by qRT-PCR. (**D**) Cells were treated with As_2_O_3_ in the presence or absence of MG132, and then protein level of Mcl-1 was confirmed by immunoblotting. *N.S*.: Non-significant. * *p* < 0.05, *** *p* < 0.005 determined from *t*-test. Data are presented as means ± SEM.

**Figure 3 cells-08-00495-f003:**
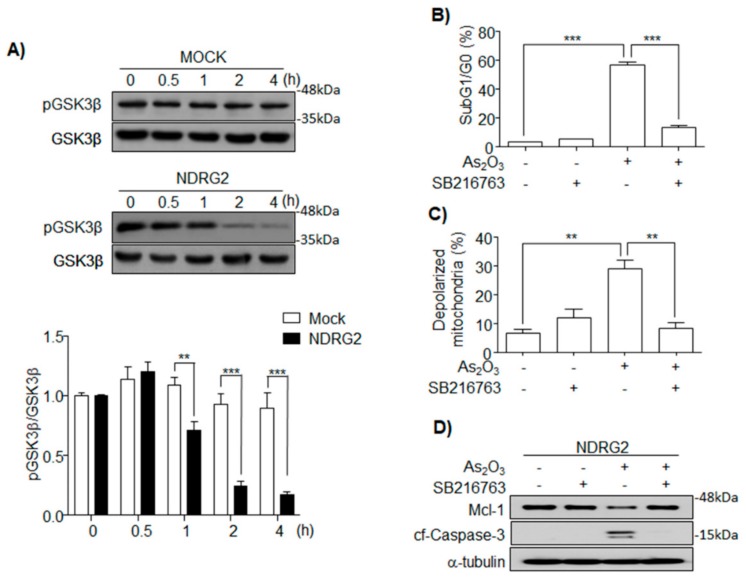
As_2_O_3_-induced glycogen synthase kinase 3β (GSK3β) activation regulates Mcl-1 degradation and apoptosis in U937-NDRG2 cells. (**A**) Kinetics of phosphorylated GSK3β (Serine 9) in U937-Mock and U937-NDRG2 was checked after treatment with 2 μM As_2_O_3_ at the indicated time. Band intensity of phosphorylated GSK3β to total GSK3β was quantified using Image J program and presented graphically. U937-NDRG2 was treated with As_2_O_3_ in the presence or absence of a GSK3β inhibitor, SB216763 (10 μM). Cell death was determined using PI staining (**B**), and mitochondrial depolarization rate was done through Mitotracker CMXRos staining (**C**). The level of the indicated proteins was analyzed using immunoblotting (**D**). ** *p* < 0.01, *** *p* < 0.005 determined from *t*-test. Data are presented as means ± SEM.

**Figure 4 cells-08-00495-f004:**
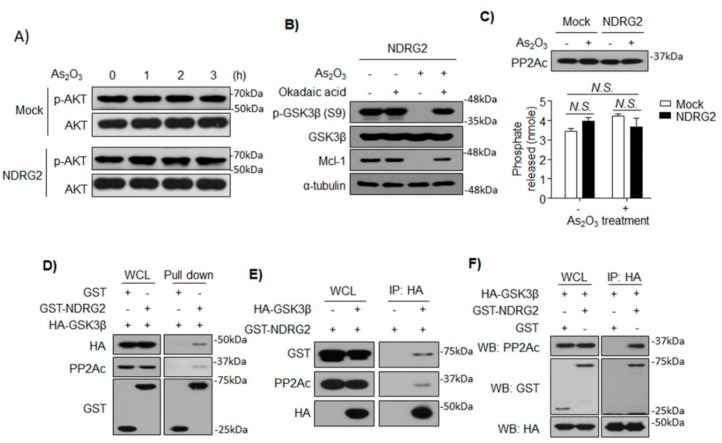
GSK3β activation observed in U937-NDRG2 is mediated by phosphatase, PP2A, not by its upstream kinase AKT. (**A**) The phosphorylation of AKT on Thr308 was determined by immunoblotting using cell lysates of As_2_O_3_-treated U937-Mock and U937-NDRG2 cells at the indicated time points. (**B**) U937-NDRG2 cells were treated with As_2_O_3_ in the presence or absence of a phosphatase inhibitor, okadaic acid (20 nM). Protein levels of phosphorylated GSK3β and Mcl-1 were analyzed by immunoblotting. (**C**) PP2A activity acquired from U937-Mock or U937-NDRG2 treated with As_2_O_3_ was analyzed with Human/Mouse/Rat Total PP2A DuoSet IC ELISA. For the quantification of PP2Ac protein level, total lysate was subjected to immunoblotting. *N.S.* no significance determined from *t*-test. Data are presented as means ± SEM. (**D**) GST or GST–NDRG2 with HA–GSK3β was co-transfected in HEK293. Protein lysates acquired from the cells were subjected to GST pull-down assay. Whole-cell lysate (WCL) was used for loading control. (**E**) Mock or HA–GSK3β with GST–NDRG2 was co-expressed in HEK293. Protein lysates acquired from the cells were subjected to immunoprecipitation with HA antibody and then interaction was confirmed by immunoblotting. (**F**) Protein lysates from HEK293 expressing HA–GSK3β with GST or GST–NDRG2 was immunoprecipitated by HA antibody. Interactions among the indicated proteins were analyzed by immunoblotting.

**Figure 5 cells-08-00495-f005:**
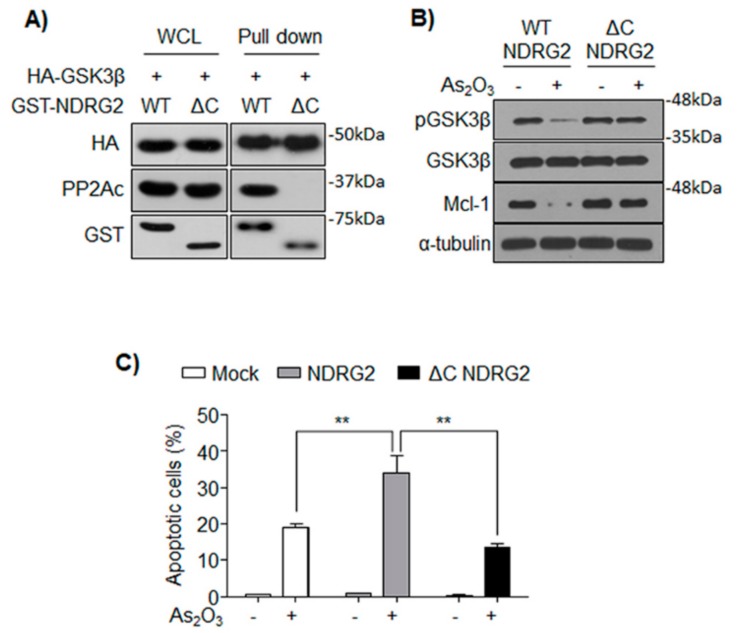
NDRG2 interacts with PP2Ac through its C-terminal, and NDRG2 ΔC is incapable of sensitizing U937-NDRG2 to As_2_O_3_. (**A**) NDRG2 wild type (WT) or NDRG2 ΔC was co-expressed with HA–GSK3β in HEK293. Prepared protein lysates were subjected to GST pull-down assay and the interaction was confirmed by immunoblotting. (**B**) U937-NDRG2 WT and U937-NDRG2 ΔC cells were treated with As_2_O_3_. Expressions of the indicated proteins were analyzed by immunoblotting. (**C**) U937-Mock, U937-NDRG2, and U937-NDRG2 ΔC were incubated with 2 μM As_2_O_3_ for 24 h. The apoptotic rate was analyzed by Annexin V/PI staining. ** *p* < 0.01 determined using *t*-test. Data are presented as means ± SEM.

**Figure 6 cells-08-00495-f006:**
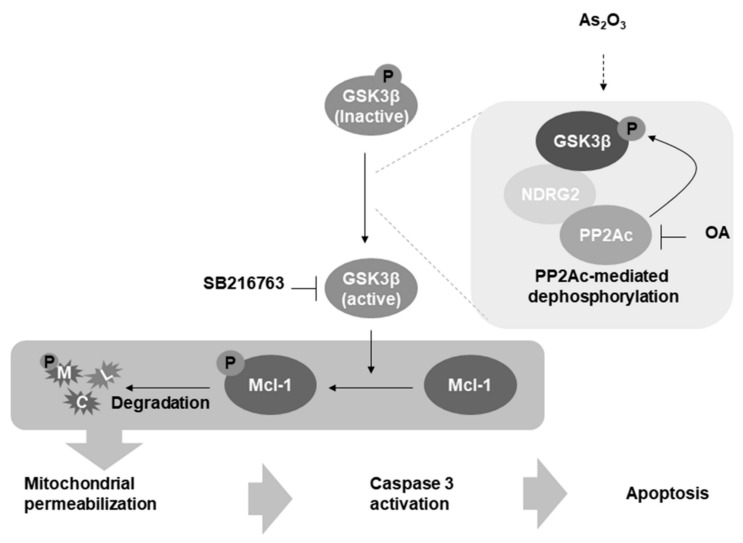
Diagrammatic representation of the proposed apoptotic pathway that mediates the higher sensitivity to As_2_O_3_ in NDRG2-expressing U937 cells.

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
