# Peer review of "NDRG2 Sensitizes Myeloid Leukemia to Arsenic Trioxide via GSK3β–NDRG2–PP2A Complex Formation"

_cells, 2019, doi:10.3390/cells8050495_

Round 1

Reviewer 1 Report

In this manuscript, Park, et al. outline interactions between GSK3b, NDRG2 and PP2A, which lead to the degradation of MCL1 following treatment with As2O3.   The authors should be commended on their very clear results with excellent figure panels. I do think, however, further discussion and explanation is needed to aid in the interpretation of these results.

Major Comments:

1.     Introduction is very brief.  The manuscript would benefit from further detail to be included in this section.  For example, where the authors are providing examples, it is useful to provide the details of what cell line was used etc. In addition, introduction switches from present to past tense, please be consistent.

2.     The generation of the NDRG2-overexpressing U937 cells has not been described in this manuscript. Please include in Methods section or reference previous publications. The fact that U937 cells do not express any endogenous NDRG2 is not mentioned until the discussion section- please include this information earlier in the manuscript as it is critical to the reader’s understanding of the model.

3.     In figure 2a, loss of MCL1 following treatment with As2O3 in NDRG2 over-expressing cells is shown.  However, these cells still express high levels of other pro-survival factors: BCL-2 and BCL-xL.  Please discuss an hypothesis as to how these cells are undergoing apoptosis following As2O3.  Are there any changes in the expression of BIM, or other pro-apoptotic factors, that could explain this synergy? Please provide immunoblots for pro-apoptotic factors.

4.     How does NDRG2 facilitate activation of GSK3b only in the presence of As2O3, given that the GSK3b/ NDRG2/ PP2A complex still forms and PP2A is not directly activated in the presence of As2O3? The authors briefly touch on this in the discussion, but further explanation is needed.

5.     In cells that are sensitive to As2O3, and express endogenous NDRG2, can the authors demonstrate that loss of NDRG2 leads to reduced sensitivity to As2O3?

Minor Comments:

1.     Please explain TNM stage and the relevance of this and EMT on the U937 tumor model.

2.     Certain batches of U937 cells from ATCC are reportedly contaminated with K562 cells. Please specify which batch number of cells was obtained from ATCC and if any cell line verification has been performed.

3.     There are some grammatical errors in this manuscript- please review thoroughly.

4.     Graphs in figure 1 are labeled with either Early apoptosis (%) or Apoptotic cell death (%), what is the difference here?

5.     Please specify what clone/ clones were used for all experiments and the timing for each treatment.

Author Response

Response to Reviewer 1 Comments

Major Comments:

Point 1.  Introduction is very brief.  The manuscript would benefit from further detail to be included in this section.  For example, where the authors are providing examples, it is useful to provide the details of what cell line was used etc. In addition, introduction switches from present to past tense, please be consistent.

Response 1. Thank you for your comment. At first, according to your comment, we changed sentences from present to past tense and added the following sentences in the Introduction part. “It was known that Mcl-1 played a role in protecting cells from apoptosis in APL cells [19]. PI3K/AKT/mTOR pathway was known to promote cell survival through translational control of Mcl-1 in acute myeloid leukemia (AML) [20]. Mcl-1 has a short half-life due to rapid degradation after phosphorylation by GSK3b which is negatively regulated by active AKT [21]. Actually, Mcl-1 protein level was decreased in As2O3-treated NB4 cells, As2O3-sensitive APL, and the decrease was dependent on GSK3b activation [22]. Therefore, strategies for AKT inhibition, GSK3b activation and degradation of Mcl-1 could be important points to improve the sensitivity to As2O3 in APL.”

Point 2-1.     The generation of the NDRG2-overexpressing U937 cells has not been described in this manuscript. Please include in Methods section or reference previous publications.

Response 2-1. We added a reference about that in 2.1 of Materials and Methods

Point 2-2. The fact that U937 cells do not express any endogenous NDRG2 is not mentioned until the discussion section- please include this information earlier in the manuscript as it is critical to the reader’s understanding of the model.

Response 2-2.  We added the follow sentence in introduction part. “In this study, we used U937 cells, which do not express NDRG2, to investigate whether NDRG2 play a role in cancer drug-sensitivity.”

Point 3.     In figure 2a, loss of MCL1 following treatment with As2O3 in NDRG2 over-expressing cells is shown.  However, these cells still express high levels of other pro-survival factors: BCL-2 and BCL-xL.  Please discuss an hypothesis as to how these cells are undergoing apoptosis following As2O3.  Are there any changes in the expression of BIM, or other pro-apoptotic factors, that could explain this synergy? Please provide immunoblots for pro-apoptotic factors.

Response 3. Actually, it was known that the downregulation of Mcl-1 through GSK-3beta activation contributed to As2O3-induced apoptosis in acute myeloid leukemia. Unfortunately, we have not available anti-Bim antibody and must order it to do the experiment you suggested. We must finish this revision within 10 days. I hope you understand this situation with good will.

Point 4.     How does NDRG2 facilitate activation of GSK3b only in the presence of As2O3, given that the GSK3b/ NDRG2/ PP2A complex still forms and PP2A is not directly activated in the presence of As2O3? The authors briefly touch on this in the discussion, but further explanation is needed.

Response 4. Thank you for your critical point. As your comment, the function of a complex seems to work only in the treatment of As2O3. Unfortunately, we could not find the reason. A reference suggested that ATO activated PP2A/C activity by downregulating miR-520h (Ann Surg Oncol. 2014 Dec;21 Suppl 4:S687-95). But the paper also did not suggest the exact mechanism to activate PP2A. We added the following sentence in Discussion part. “Although NDRG2-mediated complex formation, GSK3b/NDRG2/PP2A, could not activate PP2A directly, it might be possible that As2O3 activates PP2A [40] and then accelerates activation of GSK3b through the complex.”

Point 5.     In cells that are sensitive to As2O3, and express endogenous NDRG2, can the authors demonstrate that loss of NDRG2 leads to reduced sensitivity to As2O3?

Response 5.  We think this question is critical. When we analyzed NDRG2-mdiated sensitivity to As2O3 in several solid tumor-derived cell lines, we could not find any change in sensitivity to As2O3. Now we do not know why NDRG2 only worked in Leukemia cell line, U937. It is a possibility that Mcl-1 might play more important role to regulate apoptosis in leukemia cell compared with other solid tumor derived cells. We added a sentence, “Mcl-1 is known as a crucial component in As2O3-induced apoptosis through GSK3b activation in acute myeloid leukemia [31,36]. “ in Discussion part.

Minor Comments:

Point 1. Please explain TNM stage and the relevance of this and EMT on the U937 tumor model.

Response 1. Unfortunately, we can not find any information about that. Please understand that we can not describe about that.

Point 2. Certain batches of U937 cells from ATCC are reportedly contaminated with K562 cells. Please specify which batch number of cells was obtained from ATCC and if any cell line verification has been performed.

Response 2. Thank you for giving the important information. As your comment, ATCC comments that U937 was contaminated with 0.6% K-562 in the earliest stock. But ATCC also comments that a stock of CRL-1593 found to be free of K-562 was propagated continuously for 8 weeks and tested weekly by PCR. We purchased ATCC CRL-1593.2 and already published a research paper (Biochem Biophys Res Commun. 2010 Jun 4;396(3):684-90). And the reference was added in Materials and Methods part.

Point 3. There are some grammatical errors in this manuscript- please review thoroughly.

Response 3. We overviewed this manuscript carefully.

Point 4.  Graphs in figure 1 are labeled with either Early apoptosis (%) or Apoptotic cell death (%), what is the difference here?

Response 4. We exchanged the label “Early apoptosis (%)” with “Apoptotic cell death (%)” in Figure 1 a.

Point 5.   Please specify what clone/ clones were used for all experiments and the timing for each treatment.

Response 5. Thank you your kindness. A sentence was added in the Results 3.1. “U937-NDRG2 clone 1 was used in all experiments below”

Reviewer 2 Report

In this paper the authors demonstrated that NDRG2-overexpressing U937 cell (U937- NDRG2) showed a higher sensitivity to As2Ocompared with Mock-control U937 cell (U937-Mock). The higher sensitivity to As2Oin U937-NDRG2 was associated with Mcl-1 degradation through GSK3activation. The authors suggest that NDRG2 may be a kind of adapter protein mediating interaction between GSK3and PP2A, inducing GSK3activation which increase sensitivity to As2Oin U937 cell. 

Minors corrections

In Figure 1 c the authors affirm that NDRG2 overexpression sensitized U937 to As2Oin a caspase-dependent manner, but in the figure the absence of zVad-fmk treatment inhibit apoptosis. This must be correct.  

Author Response

Response to Reviewer 2 Comments

Thank you very much for permitting us to provide additional information during the review process of our manuscript (Manuscript ID 505059) entitled “NDRG2 sensitizes myeloid leukemia to arsenic trioxide via GSK3b-NDRG2-PP2A complex formation". We appreciate your thoughtful comments and advice.

Minors corrections

Point 1. In Figure 1 c the authors affirm that NDRG2 overexpression sensitized U937 to As2O3 in a caspase-dependent manner, but in the figure the absence of zVad-fmk treatment inhibit apoptosis. This must be correct.  

Response 1. Thank you so much for your kindness. We checked the figure.

Round 2

Reviewer 1 Report

Authors have addressed each comment to my satisfaction.

Many Thanks.